# Inhibition of Angiogenesis and Effect on Inflammatory Bowel Disease of Ginsenoside Rg3-Loaded Thermosensitive Hydrogel

**DOI:** 10.3390/pharmaceutics16101243

**Published:** 2024-09-25

**Authors:** Yiqiong Xie, Ying Ma, Lu Xu, Hongwen Liu, Weihong Ge, Baojuan Wu, Hongjue Duan, Hongmei Zhang, Yuping Fu, Hang Xu, Yuxiang Sun, Zhou Han, Yun Zhu

**Affiliations:** 1Department of Pharmacy, Nanjing Drum Tower Hospital Clinical College of Traditional Chinese and Western Medicine, Nanjing University of Chinese Medicine, Nanjing 210008, China; xieyq12580@163.com (Y.X.); xl_1006@126.com (L.X.); glg6221230@163.com (W.G.); 2Jiangsu Institute for Food and Drug Control, Nanjing 210008, China; mayingwsl@163.com; 3Department of Gastroenterology, Nanjing Drum Tower Hospital, Affiliated Hospital of Medical School, Nanjing University, Nanjing 210008, China; hongwenliu2020@163.com; 4Nanjing Medical Center for Clinical Pharmacy, Nanjing 210008, China; dhj3320092184cpu@163.com; 5Division of Breast Surgery, Department of General Surgery, Nanjing Drum Tower Hospital, Affiliated Hospital of Medical School, Nanjing University, Nanjing 210008, China; 18259721231@163.com; 6Division of Breast Surgery, Department of General Surgery, Nanjing Drum Tower Hospital Clinical College of Traditional Chinese and Western Medicine, Nanjing University of Chinese Medicine, Nanjing 210008, China; homizyjs@163.com (H.Z.); 20211683@njucm.edu.cn (Y.F.); 7School of Pharmacy, Faculty of Medicine, Macau University of Science and Technology, Macau SAR, China; njglyyxh@126.com; 8Department of Pharmacy, Nanjing Drum Tower Hospital, Affiliated Hospital of Nanjing University Medical School, Nanjing 210008, China; 9Institute of Translational Medicine, Medical College, Yangzhou University, Yangzhou 225001, China; sunyuxiang@yzu.edu.cn

**Keywords:** thermosensitive hydrogel, ginsenoside Rg3, angiogenesis, inflammatory bowel disease, drug delivery system

## Abstract

**Background**: Inflammatory bowel disease (IBD), characterized by chronic inflammation of the digestive tract, involves angiogenesis as a key pathogenic mechanism. Ginsenoside Rg3, derived from the traditional Chinese herb ginseng, is recognized for its anti-angiogenic properties but is limited by low oral bioavailability. This necessitates the development of an alternative delivery system to improve its therapeutic effectiveness. **Methods**: Pluronic F-127 (F127) and Pluronic F-68 (F68) were used to construct Rg3-loaded thermosensitive hydrogel Gel-Rg3. Meanwhile, a series of physicochemical properties were determined. Then the safety and pharmacological activity of Gel-Rg3 were evaluated in vitro and in vivo using human umbilical vein endothelial cells (HUVECs) and colitis mouse model, in order to initially validate the potential of Gel-Rg3 for the treatment of IBD. **Results**: We engineered a rectally administrable, thermosensitive Gel-Rg3 hydrogel using F127 and F68, which forms at body temperature, enhancing Rg3’s intestinal retention and slowly releasing the drug. In vitro, Gel-Rg3 demonstrated superior anti-angiogenic activity by inhibiting HUVEC proliferation, migration, and tube formation. It also proved safer and better suited for IBD’s delicate intestinal environment than unformulated Rg3. In vivo assessments confirmed increased intestinal adhesion and anti-angiogenic efficacy. **Conclusions**: The Gel-Rg3 hydrogel shows promise for IBD therapy by effectively inhibiting angiogenesis via rectal delivery, overcoming Rg3’s bioavailability limitations with improved safety and efficacy. This study provides new inspiration and data support for the design of treatment strategies for IBD.

## 1. Introduction

Angiogenesis is the process by which new capillaries are created from existing vessels; it generally involves increased vasopermeability, the degradation of the basal membrane, the migration and proliferation of vascular endothelial cells, tubulogenesis and lumen formation [1,2]. Angiogenesis is closely associated with inflammation. During chronic inflammation, the proliferation and infiltration of stromal cells create a hypoxic microenvironment, which induces the secretion of pro-angiogenic and pro-inflammatory factors by multiple cell types in inflamed tissues, driving angiogenesis. Notably, newly formed blood vessels are immature in the context of chronic inflammation; they have high vascular permeability as well as leakage, which often leads to the exacerbation of local inflammation and tissue damage [3,4]. Meanwhile, neovascularization and elevated vascular permeability increase the delivery of oxygen and nutrients to the inflammatory site, contributing to the further recruitment and soakage of inflammatory cell and immune cells [5]. Angiogenesis and inflammation thus form an inter-linked and mutually reinforcing process.

As a type of chronic inflammation, the specific pathogenesis of inflammatory bowel disease (IBD) has not been clearly defined, and has been jointly regulated by multiple factors [6,7,8,9]. Previously, researchers have explored the intrinsic link between angiogenesis and IBD [5,10,11]. In studies of mucosal tissues and serums from patients with active IBD, increased microvascular density and the elevated expression of pro-angiogenic related factors (such as VEGF, b-FGF, Ang1/2, ICAM-1, MMPs, etc.) were observed [12,13,14,15,16,17]. This indicates that angiogenesis is likely to be a vital procedure of IBD pathogenesis and a crucial link in the progression of IBD. So, it is feasible to prevent and treat IBD from the perspective of anti-angiogenesis.

Ginsenoside Rg3 (Rg3) is one of active ingredients in Chinese medicinal ginseng; it has been reported to possess multiple pharmacological effects, such as anti-oxidation, anti-inflammatory, anti-angiogenesis, anti-tumor and immune enhancement, and has been approved for health food [18,19,20,21]. Existing studies demonstrated the superior effects of Rg3 in anti-tumor angiogenesis, which is able to regulate the secretion of pro-angiogenic factors in the tumor microenvironment while affecting proliferation activity of vascular endothelial cells by acting on the G2 phase [22]. In our preliminary literature research, we found that many researchers have explored the anti-inflammatory effect of Rg3 in the treatment of IBD, while the regulatory effect of Rg3 on angiogenesis in IBD has not been investigated [23,24]. However, Rg3 has low solubility, resulting in poor gastrointestinal absorption and low oral bioavailability [25,26]. To address the problem, preparations were used to improve the utilization of Rg3, such as liposomes, electrospun membranes, etc., so it can consequently play a better role in disease treatment [27,28,29].

Hydrogels have been applied in the research and development of wound dressing and in situ injections due to their excellent biocompatibility, biodegradability, local adhesion and drug-loaded internal structure with reticular cross-linking [30,31,32,33,34]. In the intestinal administration of IBD therapy, the rapid release of drugs may lead to further stimulation of the fragile intestinal mucosa. Therefore, appropriate dosage forms must include loading drugs to a certain extent to ensure a slow release for the necessary protection of intestinal mucosa; hydrogel is an ideal dosage form with good injectivity and a certain sustained-release effect [35]. Poloxamer is a common material of gels and is FDA-approved as an inactive ingredients for use in drug delivery systems [36]. The hydrogel prepared with Pluronic F-27 (F127) and Pluronic F-68 (F68) has temperature-sensitive properties. With the change in F127/F68 ratio (*w*/*w*), it can transform into a gelation state at different temperatures, which helps us to accomplish drug delivery via the characteristic of temperature dependence. Furthermore, the interaction between the Pluronic hydrogel and mucins within the colonic mucosal tissue, coupled with tissue dehydration induced by enhanced water absorption and hydrogel swelling subsequent to gelation within the intestinal milieu, collectively govern the adhesive properties of the hydrogel within the intestinal environment [37,38,39]. Hydrogels undergo gelation and adhere to the gut; with the degradation of the gel in vivo, the drug is slowly released, thus achieving the aim of reducing drug leakage, increasing the duration of drug action and improving the bioavailability of drugs [40].

In our study, we aimed to develop and evaluate a novel thermosensitive hydrogel formulation of Ginsenoside Rg3 (Gel-Rg3), designed to leverage its anti-angiogenic properties for the treatment of inflammatory bowel disease (IBD). Our hypothesis was that Gel-Rg3, which transitions from a liquid to a solid state at body temperature (35–37 °C), would enhance drug solubility, provide sustained release and improve therapeutic efficacy and safety compared to conventional treatments. To achieve this, we formulated the hydrogel to adhere within the 37 °C intestinal environment [40], evaluated its physico-chemical properties, tested its safety and pharmacological activity in vitro using human umbilical vein endothelial cells (HUVECs) and assessed its retention capacity and efficacy in vivo by inhibiting angiogenesis. The significance of this study lies in its potential to offer a more effective and safer therapeutic option for IBD by advancing the delivery and effectiveness of Rg3. The schematic representation of our hypothesis is shown below (Figure 1).

## 2. Materials and Methods

### 2.1. Materials

Ginsenoside Rg3 (Rg3), Pluronic F-27 (F127) and Pluronic F-68 (F68) were purchased from Yuanye Biotechnology (Shanghai, China). Dulbecco’s modified Eagle’s medium (DMEM), trypsin, fetal bovine serum (FBS) and phosphate-buffer solution (PBS) were supplied by BioChannel Biotechnology (Nanjing, China). Dimethyl sulfoxide (DMSO) was obtained from Sinopharm Chemical Reagent (Shanghai, China). The Cell Counting Kit-8 (CCK-8) was purchased from Vazyme Biotech (Nanjing, China). The Calcein AM Cell Viability Assay Kit was purchased from Beyotime Biotechnology (Shanghai, China). Propidium iodide dye liquor was purchased from Keygen Biotech (Nanjing, China). Matrigel was supplied by Corning Incorporated (New York, NY, USA). Indocyanine Green (ICG) was purchased from Aladdin (Shanghai, China). Dextran sodium sulfate (DSS, MW 36,000–50,000) was purchased from MP Biomedicals (Irvine, CA, USA). Isoflurane was purchased from Beijing Orbiepharm (Beijing, China).

### 2.2. Cell Culture

The human umbilical vein endothelial cell line (HUVECs) was obtained from Laboratory of Gastroenterology, Nanjing Drum Tower Hospital. The cells were cultured in Dulbecco’s modified Eagle’s medium (DMEM) supplemented with 10% fetal bovine serum (FBS) and 1% penicillin/streptomycin. And the cells were incubated in a humidified atmosphere at 37 °C and 5% CO_2_.

### 2.3. Animals

Male C57 BL/6 J mice aged 5–6 weeks and weighing 18–20 g were purchased from Cavens Biotechnology (Nanjing, China) and acclimated in the laboratory for 1 week. All animal experiments were approved by the Ethics Committee and Animal Welfare Committee of Nanjing Hospital Affiliated to Nanjing University Medicine School (DWSY-23093397).

All the experimental groups were formed via randomization, and all the mice were randomly numbered, divided into groups in numerical order and randomly assigned to experimental or control groups. A gas anesthesia apparatus (Matrx VIP3000, Midmark, FL, USA) was used to anesthetize the mice via inhalation (3% isoflurane balanced with oxygen) before each rectal administration in order to reduce discomfort and avoid the stressful state of the mice during the administration. All mice were euthanized using cervical dislocation at the end of the experiment.

### 2.4. Fabrication of Gel-Blank and Gel-Rg3

The F127/F68 blank hydrogel was prepared using the cold solution method. First, 0.25 g F127 and 0.125 g F68 were weighed according to prescription and added to 1 mL deionized water. The formed mixture was placed in the refrigerator at 4 °C overnight to make it fully dissolve. Thus, the Gel–Blank hydrogels were obtained. The preparation of other formulation ratios can be adjusted based on this method.

The Gel-Rg3 hydrogel was prepared using the same method. However, one key difference was that the deionized water was replaced by an Rg3 solution of different concentrations, which was dissolved in DMSO aqueous solution (10%, *v*/*v*).

### 2.5. Determination of Transition Temperatures (T_sol-gel_)

Sample hydrogels (1 mL) with different F127/F68 proportions were prepared in accordance with the fabrication method we mentioned in Section 2.4 in clear Eppendorf tubes, and placed on a magnetic rotor before measurement. Then, the sample tube was dipped into a magnetic stirring water bath at 30 °C and heated at a speed of 1–2 °C/min, while turning on the magnetic stirring. When the magnetic rotor stopped rotating due to gelation, the temperature displayed by the thermometer was recorded as the solution–gelation transition temperature (T_sol-gel_).

### 2.6. Characterization of Gel-Rg3

All hydrogel samples were lyophilized before structural testing. The morphologies of Gel-Blank and Gel-Rg3 were examined by using scanning electron microscopy (JSM IT200, Jeol, Tokyo, Japan). The freeze-dried hydrogels were sputter-coated with gold before detection, and then imaged using SEM at different magnifications to observe the microstructure. The chemical groups of Pluronic F-127, Pluronic F-68, Ginsenoside Rg3, Gel-Blank and Gel-Rg3 were identified by using Fourier transform infrared spectroscopy (FT-IR) (Nicolet iS50, ThermoFisher, Waltham, MA, USA). The freeze-dried Gel-Blank and Gel-Rg3, as well as the powders of F127, F68 and Rg3 were mixed with potassium bromide and ground into ultrafine particles. Next, the mixed powders were compressed into a flake and scanned using FT–IR.

### 2.7. Rheology Research

Rheological properties of Gel-Blank and Gel-Rg3 were measured using a rheometer (DHR-2, TA). The strain and frequency were constantly set as 0.05% and 1 Hz to determine the change in the storage modulus (G′) and loss modulus (G″) of hydrogels in the temperature of 0–45 °C, and the value of the loss factor tanδ was calculated. Meanwhile, the change in the gel kinetic viscosity with increasing shear rate was determined at a constant temperature (25 °C).

### 2.8. Release Test of Gel-Rg3

Gelatinized Gel-Rg3 (500 μL) was placed into simulated colonic fluid (15 mL) (0.01 M PBS, pH 7.4, with 0.05% SDS) which was preheated at 37 °C. Then, the samples were transferred to a 37 °C homothermal shaking bath and 1 mL of the samples was withdrawn at a predetermined time (0.5, 1, 2, 4, 6, 8, 12, 24, 48 and 72 h) and replaced by equal amounts of fresh medium. The concentration of Rg3 was determined by using high performance liquid chromatography (HPLC) analysis (Waters 2695, Waters, MA, USA) at the Jiangsu Institute for Food and Drug Control according to standard methodology. The specific chromatographic conditions were as follows: chromatographic column—Agilent ZORBAX SB; C18 column (4.6 × 250 mm, 5 μm) (Agilent, Santa Clara, CA, USA); mobile phase—acetonitrile/water (48/52); flow rate—1 mL/min; detection wavelength—203 nm; injection volume—20 μL; column temperature—25 °C. Each experiment was repeated in triplicate. The drug release rate was calculated at each time point and fitted to the curve.

### 2.9. In Vitro Cytotoxicity

Live/dead cell staining was used to evaluate the in vitro safety of Gel-Rg3 using a Calcein AM Cell Viability Assay Kit and propidium iodide dye liquor (PI). Human umbilical vein endothelial cells (HUVECs) were cultured in 12-well plates at a density of 30,000 cells per well. After an overnight incubation, cells were treated with Gel-Rg3-leaching solution, Rg3 or DMSO aqueous solution (10%, *v*/*v*) for 72 h. At the end of treatment, Calcein AM working solution was applied to live cell staining in green fluorescence, and PI working solution was applied to dead cell staining in red fluorescence, as per the recommendations mentioned in the product manuals. And then, live and dead cells were observed under a fluorescence microscope (EVOS M7000, ThermoFisher).

### 2.10. In Vitro Cell Proliferation

Cell proliferation was measured using the CCK-8 kit. HUVECs (5000 cells per well) were inoculated in 96-well plates; each group was provided with three multiple repeat holes. After a 24 h incubation, the medium was replaced with different concentrations of Gel-Blank (5, 10, 20%, *v*/*v*), Rg3 (20, 50, 100, 200 μg/mL) and Gel-Rg3-leaching solution (20, 50, 100, 200 μg/mL), and the control group was given a certain amount of solvent (DMSO aqueous solution, 10%, *v*/*v*). When the incubations for 24, 48 or 72 h were finished, the CCK-8 reagent was added to each well and incubated for 0.5–2 h at 37 °C, and the absorbance was measured at 450 nm with a microplate reader (Spark 20 M, Tecon, Singapore).
$$Inhibition\;rate(\%)=\frac{ODcontrol-ODsample}{ODcontrol}\times 100\%$$

### 2.11. In Vitro Scratch Assay

HUVECs were seeded onto 6-well plates at a density of 500,000 cells per well and cultured for 24 h to obtain a monolayer. An amicrobic 200 μL pipette tip was used to create straight scratches, and the fallen cells were removed by rinsing with PBS. Then, the cells were co-incubated with or without treatment of Rg3 (50 μg/mL) and Gel-Rg3-leaching solution (20, 50 μg/mL) in serum-free DMEM. Cell migration phenomena were photographed at 0, 24, 48, 72 h with an inverted microscope. The measurements of blank area (S_x_) were performed using Image J (Java 1.8.0_345).
$$Cell\;migration\;rate(\%)=\frac{S_{initial}-S_{final}}{S_{final}}\times 100\%$$

### 2.12. Tube Formation Experiments

A matrigel-based assay was used to observe the formation of capillary tube-like structures after treatment. In total, 10,000 cells per well of HUVECs were cultured in suspension with DMEM only or containing various doses of Rg3 (50, 100, 200 μg/mL) and Gel-Rg3 leaching solution (50, 100, 200 μg/mL) in a 48-well plate prelaid with matrigel. The formation tube-like structures were photographed at 8 h using an EVOS M7000 imaging system.

### 2.13. In Vivo Retention

Mice were randomly divided into two groups; each group rectally received 50 μL of free fluorescent probe ICG (2 mg/kg) or our hydrogels loaded with ICG (2 mg/kg). At 2 h and 10 h after the administration, the dissected colon tissues were photographed using an in vivo imaging system (IVIS Lumina XR, Caliper, Life Sciences, Waltham, MA, USA) to observe the retention of drugs in intestinal tract. The tissues were subsequently fixed with 4% paraformaldehyde, and the fluorescence intensity of the intestinal retained drugs was observed via fluorescence microscopy (Thunder, Leika, Wetzlar, Germany) after frozen sectioning (*n* = 3). The semiquantitative analysis of the mean fluorescence intensity was performed using Image J.

### 2.14. In Vivo Efficacy Evaluation

The mice were randomly assigned to four groups (*n* = 3): (1) Normal, (2) Model, (3) DSS + Rg3, (4) DSS + Gel-Rg3. All mice were given 2.5% DSS solution (*w*/*v*) for 4 days to establish an IBD model, except for the normal group of mice that were allowed to drink normal water. When the mice developed looser stools and blood stools, the three groups of molded mice were given an equal volume of saline, Rg3 and Gel-Rg3 via rectal administration once every two days for a total of four administrations. The rectal administration of the hydrogel was achieved with the help of a 1 mL syringe and a 1 mm diameter rubber hose, at a dose of 50 μL per mouse per dose (with 0.25 g/mL F127, 0.125 g/mL F68 and 30 mg/kg Rg3), with a 12 h fasting period before each administration [41,42]. After the final administration, the mice were euthanized and the colon length was measured; the distal colon tissues were fixed in a 4% paraformaldehyde solution for hematoxylin and eosin (H&E), immunohistochemical (IHC) and immunofluorescence (IF) staining analysis.

### 2.15. Statistical Analysis

Quantitative data analyses and graphs were performed using GraphPad Prism 9 for Student’s *t*-test or one-way ANOVA analysis; the statistics of all experiments were expressed as mean ± SDs (*n* = 3). *p* < 0.05 was set as statistically significant.

## 3. Results and Discussion

### 3.1. Prescription and the Characterization of Thermosensitivity for Gel-Rg3

The hydrogels prepared with F127 and F68 are temperature-sensitive and can gel at a certain temperature, which is defined as the phase transition temperature of the gel (T_sol-gel_). As shown in Figure 1, T_sol-gel_ is affected by the proportion of hydrogel components; it was observed that T_sol-gel_ decreased with an increase in the ratio of F127, whereas an increase in the ratio of F68 led to its rise. According to the T_sol-gel_ curve of Gel-Blank (Figure 1A,B), 25% F127 (*w*/*v*) and 7.5% F68 (*w*/*v*) were selected for the following investigation. Subsequently, we found that the T_sol-gel_ decreased by 3–4 °C after loading Rg3, so we reset the concentration of F127 and F68 to draw the new T_sol-gel_ curve of the loaded gel (Figure 1C,D) and determined that the hydrogel prescription as 25% F127 (*w*/*v*), 12.5% F68 (*w*/*v*) and 200 μg/mL Rg3, of which the gelation time was 87.3 ± 3.1 s at 37 °C.

### 3.2. Characterization and Properties of Gel-Rg3 Hydrogels

The morphology, Fourier transform infrared spectroscopy (FT–IR) spectral and scanning electron microscopy (SEM) images results of Gel–Blank and Gel-Rg3 are shown in Figure 2. When the temperature transformed from 4 °C to 37 °C, both of the Gel–Blank and Gel-Rg3 hydrogels changed from a flowing-liquid state to a solid-gel state, and returned to a liquid state after cooling, which shows that the sol–gel transition of our hydrogels is reversible. The resulting gel solution was clear and transparent, indicating that Rg3 was well dissolved and mixed in the system. The solidified gel remained solid at 37 °C and stuck on the wall of the vial, indicating that it can commendably adhere to the gut and retain its morphology after transrectal administration (Figure 2A). The characteristic peaks of the F127, F68, Rg3, F127/F68 blank hydrogels and the Rg3-loaded hydrogel were confirmed using Fourier transform infrared spectroscopy (Figure 2B). F127, F68 and Gel–Blank showed the typical peaks at 3451 cm^−1^, 1115 cm^−1^ (allocated to the stretching and bending of O-H) and 2284 cm^−1^ (allocated to the stretching of C-H). Rg3 showed the peaks at 3409 cm^−1^, 1077 cm^−1^ (allocated to the stretching and bending of O-H), 2933 cm^−1^ (C-H stretching) and 1635 cm^−1^ (C=C stretching) [43]. We observed that the chemical structure of the Gel-Rg3 is similar to F127, F68 and Gel-Blank; the typical broad peaks of Rg3 disappeared after it was loaded into the gel, which is consistent with the previously reported characteristics of poloxamer gels [44]. This may be due to the fact that after being encapsulated, the infrared absorption of Rg3 was affected by F127 and F68 [45]. For example, as the major component in mixture, the strong absorption of F127 and F68 at certain wave numbers may cause the baseline to lift, affecting the clarity and strength of the absorption peaks of other components. It may also overlap with the absorption peaks of a drug or other component, causing the characteristic peaks of the drug or other component to be masked or weakened [46]. In addition, there may be interactions between F127 and F68 and drugs or other components, such as hydrogen bonds, van der Waals forces, etc. [43]. The SEM image illustrated an evident three-dimensional porous reticular structure of our hydrogel, based on which the loading of drugs was realizable (Figure 2C,D) [30,47].

The mechanical properties and drug release features in the intestine were evaluated through rheological tests and the drug release profile in vitro. The calibration curve of Rg3 measured using HPLC is shown in Appendix A. Concerning the release profile, as shown in Figure 2E, about 33.8% of Rg3 was rapidly released from Gel-Rg3 over 6 h, followed by a continuous slow release to 61.6% over 72 h, showing its better sustained-release performance in the intestinal environment, which is beneficial to the local accumulation of Rg3 and prolongs the drug duration of action. The rheological data of the hydrogels showed that the storage modulus (G′) and loss modulus (G″) of both Gel–Blank and Gel-Rg3 increased with the temperature, eventually leading to equilibrium (Figure 3A,B). When the temperature is below 36.2 °C, G″ is much greater than G′, indicating that the hydrogel mainly undergoes viscous deformation, resembling a liquid with good flowability. When the temperature reaches approximately 36.2 °C, the storage modulus and loss modulus are approximately equal, indicating that the drug-loaded hydrogel starts to transition from a liquid to a solid state, forming a semi-solid gel with poor fluidity [48,49]. This reflects the successful formation of gelation after reaching the phase transition temperature, further highlighting the adhesion ability of our hydrogel in the gut. Moreover, with the increasing shear rate, the viscosity of both hydrogels decreased significantly, showing good shear thinning properties, indicating that they were injectable and that rectal administration could be achieved (Figure 3C,D) [30]. These outcomes demonstrate the feasibility of our hydrogel for rectal administration. It can be injected into the gut in a liquid state at room temperature, adapting to the irregular surface of the intestine to achieve adherence to the inflamed gut. Subsequently, the hydrogel undergoes gelation within a certain period of time, facilitating its retention in the gut and then initiating sustained drug release.

### 3.3. In Vitro Cytotoxicity Evaluation

On account of the fact that the exposure of drugs at high dosage might lead to damage to a certain degree, the advantages of Gel-Rg3 with a sustained-release profile was measured in vitro. Live/dead staining was used to observe the cytotoxicity of the Gel-Rg3 and Rg3; the distributions of live and dead cells after the administration of various concentrations of drugs for 72 h are displayed in Figure 4A. As we have seen, live cells were presented with green fluorescence after Calcein AM staining, while dead cells were stained red by propidium iodide (PI) due to alterations in cell membrane permeability; a portion of the dead cells could not be shown in the picture because of shedding. A comparison of viable cell densities between groups revealed that the cytotoxicity of Rg3 was concentration-dependent, as a low concentration of Rg3 had less cytotoxicity (Figure 4A). When the concentration of Rg3 reached 50 μg/mL, the proportion of viable cells in the free Rg3 group was 31.06% while the percentage of viable cells treated with Gel-Rg3 at the same concentration still managed to reach 67.57%; even with a four-fold increase in the concentration, the Gel-Rg3 group was able to retain 22.09% of the viable cell area (Figure 4B). Compared with the free Rg3 in an equal concentration, the cell viability of the Gel-Rg3 was preferable, suggesting that the hydrogel material has superior biological safety.

### 3.4. In Vitro Cell Proliferation

To further prove the safety of our materials, we incubated the HUVECs with Gel–Blank, and the results showed that the cell viability of HUVECs after exposing to blank hydrogels with a concentration of 10% (*v*/*v*) or less was above 95%, suggesting that the hydrogel was cell-compatible in this concentration range (Figure 4C). The results from the Cell Counting Kit-8 (CCK-8) showed that different concentrations of Rg3 and Gel-Rg3 had a certain inhibitory ability on the proliferation of HUVECs compared with the control group, and this effect was gradually significant with the increase in drug concentration and the extension of administration time (Figure 4D,E). Obviously, we find that free Rg3 is more lethal to vascular endothelial cells when compared with an equal concentration of Gel-Rg3. After co-culturing with 200 μg/mL free Rg3 for 48 h, the inhibition rate of cell proliferation exceeded 98%, whereas Gel-Rg3 led to a 41.9% inhibition under the same conditions, which further demonstrated the safety of Gel-Rg3. It can play the role of inhibiting proliferation while reducing the killing effect of high concentration drugs on healthy cells, which could possibly benefit from its sustained-release characteristics [26].

### 3.5. In Vitro Anti-Angiogenesis Assay

Scratch assay was performed to probe whether Gel-Rg3 could inhibit the migration of vascular endothelial cells, which is a key link in neovascularization. In the control group, the cells reached about 80% confluence, while the groups treated with Gel-Rg3 and Rg3 showed a significant inhibitory effect of migration ability, and it was more obvious in the high-concentration group (Figure 5A). The results further indicated the anti-angiogenic capacity of Rg3. Under the light microscope, we found that the cells were in a significantly better condition in groups of Gel-Rg3 than in an equivalent concentration of free Rg3, which reflected the safety of the gel material from the side.

Neovasculogenesis plays an important role during the angiogenesis process. Our result from the tube formation experiment shows that capillary tube-like structures were effectively suppressed after cultivating with Rg3 or Gel-Rg3, and this inhibition is positively correlated with the drug concentration (Figure 5C). In the control group, we could clearly observe the formation of tube-like structures, while the number of visible vascular branches, nodes and junctions became fewer as the concentration of the administered drug increased, and tube-like structures were hardly observed at high concentrations. Especially in the 200 μg/mL Rg3 group, the cells were almost suspended independently in the medium without any tendency to form tubules. In addition, we found that this was consistent with the results of CCK-8 test and the scratch assay we presented previously, Gel-Rg3 exhibited a better safety profile at the same administered concentration, which presented a better cellular status. This may have great significance for its ability to inhibit angiogenesis while protecting the fragile intestinal mucosa from irritation when applied in reality [1].

### 3.6. The Intestinal Retention Ability of Thermosensitive Hydrogel

Based on the thermosensitive property of the hydrogel we prepared, it can transition from solution to gelation. Its rheological property, where its fluidity decreases with rising temperature, combined with the bio-adhesion property of poloxamer hydrogels, creates favorable conditions for its retention and adhesion in the gut [37,39,50,51,52,53]. To simulate the actual retention of the gel in vivo, Indocyanine Green (ICG) was encapsulated in the gel for rectal administration, and free ICG solution was used as a control. The results showed that the fluorescence of the intestine in the gel group increased with the passage of time after administration, which was superior to that in the free group, suggesting that the gel has better intestinal retention characteristics (Figure 6A). Subsequently, in order to further explore the penetration and absorption of drugs in the intestine after transrectal administration, the intestine was collected at each time point, and the distal colon was cut off for the frozen section. The fluorescence intensity of the colon tissue section in each group was observed and compared under a fluorescence microscope. Representative fluorescence images are displayed in Figure 6B; the average fluorescence intensity of colon sections in the Gel group increased with time, and its fluorescence intensity reached two times of that in the free ICG group at 10 h (Figure 6C). Similarly, the fluorescence area on the intestinal mucosa of the gel group was much larger (Figure 6D), showing a favorable intestinal retention capacity and drug absorption. Interestingly, the fluorescence intensity of colon sections in the free ICG group dropped significantly. At 10 h, it was nearly zero. This could be because the absorbed drug was partially metabolized over time. Additionally, the unabsorbed drug was likely cleared with the intestinal contents during pre-slice processing. In contrast, the gel has the characteristics of the sustained release of drugs, as the drug loaded in the gel continued to be released slowly and accumulated locally. Therefore, strong fluorescence could still be detected in the intestinal mucosa at 10 h after administration, which further suggested the advantage of our temperature-sensitive gel in rectal administration.

### 3.7. Anti-Angiogenesis Efficacy of Gel-Rg3 on IBD In Vivo

Since colon length is an important indicator of the severity of IBD, we collected the intestines of mice in each group and measured their length at the end of the treatment cycle; we found that the colon length of mice in the untreated IBD group (control group) was significantly shorter compared to the mice in the normal group, and was restored to some extent after the administration of either Rg3 or Gel-Rg3, with a significant difference compared to the control group (Figure 7A,B). Hematoxylin and eosin (H&E) staining was then used to examine colon histological and morphological features. In the normal group, colon tissue showed complete structure, clearly visible crypt morphology and abundant goblet cells, while muscle thickening, crypt loss and goblet cell injury were found in the control group [54]. These phenomena were superbly alleviated after the administration of Rg3 or Gel-Rg3, suggesting the relieving effect of Rg3 on DSS-induced colitis (Figure 7C). Especially, after Gel-Rg3 treatment, the above indexes tend to be repaired in the normal group.

To further investigate the effects of Rg3 and Gel-Rg3 on angiogenesis in IBD in vivo, we labeled the CD31 in the colonic tissue through immunostaining to evaluate the vascular density in the colonic mucosal layer [14]. The results of both immunofluorescence and immunohistochemistry showed that the vascular density in the colon was significantly elevated after DSS-induced in mice, and that the vascular density decreased back to the normal level after treatment with Rg3 or Gel-Rg3, indicating the excellent anti-angiogenic effect of Gel-Rg3 as well as of Rg3 in vivo (Figure 7D,E). The efficacy of Gel-Rg3 was still comparable to free Rg3 even under sustained-release conditions. Combined with the results of the in vivo retention assays, Gel-Rg3 has a longer-lasting efficacy and safety profile, showing the potential to be utilized in practical therapeutic applications at a much lower dosing frequency.

## 4. Conclusions

In summary, we prepared a temperature-sensitive F127/F68/Rg3 hydrogel to increase the availability of Rg3 for IBD. It can be delivered into the inflamed gut via rectal administration, adhering to the intestine with its gelling properties at 37 °C, and releasing drugs slowly with the biodegradation of the gel material. Thus, a reduction in drug leakage and a prolonged duration of drug action were achieved. Our results showed that the resulting hydrogel has a mesh cross-linked structure to pack Rg3 in it and has an excellent sol–gel convertibility. In addition, Gel-Rg3 has a certainly sustained-release effect and better biological safety than free Rg3, which reduces the stimulation of the drug to the susceptible intestinal mucosa. Gel-Rg3 showed an inhibitory effect on the proliferation and migration of HUVECs in the pharmacodynamic experiment in vitro, suggesting its anti-angiogenesis capacity. The in vivo experiment also showed a strong inhibition of angiogenesis from Gel-Rg3, which was consistent with our conclusion in vitro. In comparison to free Rg3, Gel-Rg3 was more potent, not only in terms of comparable CD31 levels, but also in terms of its superior retention and sustained-release properties at the colonic site, which underscored the higher safety profile in vivo. It is unquestionable that the superior anti-angiogenic effect of Rg3 has been reconfirmed in IBD, and the application of Rg3 to the treatment of IBD may be a valuable therapeutic idea.

## Data Availability

The data that support the findings of this study are available from the corresponding author upon reasonable request (the data are not publicly available due to privacy).

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
