# Peer review of "Inhibition of Angiogenesis and Effect on Inflammatory Bowel Disease of Ginsenoside Rg3-Loaded Thermosensitive Hydrogel"

_pharmaceutics, 2024, doi:10.3390/pharmaceutics16101243_

Round 1
Reviewer 1 Report
Comments and Suggestions for Authors
The article titled 'Ginsenoside Rg3 Loading Thermosensitive Hydrogel Mitigates Inflammatory Bowel Disease by Inhibiting Angiogenesis' by Xie and co-authors deserves to be published in Pharmaceutics after minor revision. The study explores the preparation of a thermosensitive hydrogel carrying ginsenoside Rg3 for rectal administration, an angiogenesis inhibitor, in order to improve its bioavailability and therapeutic effect in inflammatory bowel disease.
The title could be improved, as it may be difficult for readers to quickly grasp the main objective of the study. Perhaps 'Inhibition of angiogenesis and effect on Inflammatory Bowel Disease of ginsenoside Rg3-loaded thermosensitive hydrogel' would be better.
On line 13: “Hospital, the Affiliated Hospital of Medical School” should perhaps be “Hospital, Affiliated Hospital of the Medical School”.
The authors should add the keyword ‘Drug delivery systems’ to improve the visibility of the paper.
The introduction describes the aim of the study to develop a thermosensitive hydrogel for rectal administration of Rg32. However, the hypothesis could be stated more explicitly to highlight the novelty and importance of the research.
The authors should explain how they determined that the number of experiments should be 3 in all cases.
Scheme 1 is not mentioned in the text of the paper.
Some long sentences should be simplified to make it easier to read and capture the information the authors mean.
The authors mention supplementary materials in the corresponding section, but do not include them. Perhaps this is sentence in the template document that should be removed. The funding statement should also be more specific.
Comments on the Quality of English LanguageA native English speaker, or someone with sufficient experience, should review the entire document to simplify some long sentences to make the study easier to read.
Reviewer 2 Report
Comments and Suggestions for Authors
Ginsenoside Rg3 Loading Thermosensitive Hydrogel Mitigates Inflammatory Bowel Disease by Inhibiting Angiogenesis
This manuscript contains the original research work in which the authors developed a Ginsenoside Rg3 (Rg3) loading thermosensitive hydrogel for rectal administration to enhance the bioavailability of Rg3 and explore its potential in inflammatory bowel disease (IBD) therapy. The common and economical materials Pluronic F - 127 (F127) and Pluronic F - 68 (F68) were used to load Rg3 due to the internal reticular cross-linked structure of gels. The resulting Gel-Rg3 hydrogel undergoes gelation at intestinal temperature upon administration, adhering to the gut accompanied by slow release, thus reducing intestinal mucosa irritation and prolonging drug action duration. In vitro experiments showed that Gel-Rg3 exhibited favorable inhibitory effects on the proliferation, migration and tube formation of HUVECs, indicating its efficacy in anti-angiogenesis. Compared to the direct administration of free Rg3, Gel-Rg3 displayed enhanced safety and suitability for the sensitive gut of IBD patients. In addition, in vivo retention assays and pharmacodynamic evaluations confirmed the superior intestinal adhesion as well as the anti-angiogenic efficacy of Gel-Rg3. The analysis of the obtained results suggests that the developed hydrogels could be the promising candidate for inhibiting angiogenesis in IBD by rectal administration.
I have no fundamental remarks on the work. All the obtained results are carefully analyzed, and the conclusions drawn are justified. I recommend accepting the current manuscript, however there are some aspects in this manuscript that should be improved:
Comment 1: I suggest the authors rewrite the last paragraph of the Introduction section to clearly state the objectives, mention the hypothesis, and highlight the significance of the study.
Comment 2: References. Kindly elaborate more on the sub-sections of the Results and Discussion section with references. Appropriate references are required to support the claim and results of any review.
Comment 3: Page 4, line 148: “F127 147 and F68 were weighed according to prescription and added to deionized water….” Please give more detail for the preparation of Gel - Blank and Gel-Rg3.
Comment 4: Please include the calibration curve of the HPLC method and the concentration range for Rg3.
Comment 5: Page 7, line 285: Explain in the text why he typical broad peaks of Rg3 disappeared after it is loaded into the gel.
Comment 6: Figure 2: It is an impressive exercise in synthesizing all the characterization of the hydrogels, but it has the disadvantage that the FTIR graph is not clear, could you please enlarge this graph?
Comment 7: Figures: kindly use a large font while making images.
Comment 8: Please, write in vivo, in vitro with cursive letters when used throughout the text.
